# Rapid discovery of self-assembling peptides with one-bead one-compound peptide library

Pei-Pei Yang[1,7], Yi-Jing Li[1,2,7], Yan Cao [3,7], Lu Zhang [4], Jia-Qi Wang[1], Ziwei Lai[3], Kuo Zhang [1,2], Diedra Shorty[4], Wenwu Xiao[4], Hui Cao[2], Lei Wang [1✉], Hao Wang[1,5✉], Ruiwu Liu[4✉] & Kit S. Lam [4,6✉]

Self-assembling peptides have shown tremendous potential in the fields of material sciences, nanoscience, and medicine. Because of the vast combinatorial space of even short peptides, identification of self-assembling sequences remains a challenge. Herein, we develop an experimental method to rapidly screen a huge array of peptide sequences for self-assembling property, using the one-bead one-compound (OBOC) combinatorial library method. In this approach, peptides on beads are N-terminally capped with nitro-1,2,3-benzoxadiazole, a hydrophobicity-sensitive fluorescence molecule. Beads displaying self-assembling peptides would fluoresce under aqueous environment. Using this approach, we identify eight penta-peptides, all of which are able to self-assemble into nanoparticles or nanofibers. Some of them are able to interact with and are taken up efficiently by HeLa cells. Intracellular distribution varied among these non-toxic peptidic nanoparticles. This simple screening strategy has enabled rapid identification of self-assembling peptides suitable for the development of nanostructures for various biomedical and material applications.

[1] CAS Center for Excellence in Nanoscience, CAS Key Laboratory for Biomedical Effects of Nanomaterials and Nanosafety, National Center for Nanoscience and Technology (NCNST) No. 11 Beiyitiao, Zhongguancun, Beijing, China. [2] Department of Materials Physics and Chemistry, School of Materials Science and Engineering, University of Science and Technology Beijing, Beijing, China. [3] Institute for Advanced Study, Shenzhen University, Guangdong, China. [4] Department of Biochemistry and Molecular Medicine, UC Davis NCI-designated Comprehensive Cancer Center, University of California Davis, Sacramento, CA, USA. [5] Center of Materials Science and Optoelectronics Engineering, University of Chinese Academy of Sciences, Beijing, China. [6] Division of Hematology and Oncology, Department of Internal Medicine, School of Medicine, University of California Davis, Sacramento, CA, USA. [7] These authors contributed equally: Pei-Pei Yang, Yi-Jing Li, and Yan Cao. ✉email: wanglei@nanoctr.cn; wanghao@nanoctr.cn; rwliu@ucdavis.edu; kslam@ucdavis.edu

A total of 20 eukaryotic amino acids allows the creation of tens of thousands of proteins with a stable three-dimensional structure and a variety of functionalities. Although short linear peptides (5-mer to 10-mer) lack a stable three-dimensional structure, some of them may have a stable secondary structure. Others may possess self-assembling properties. It is well documented that the shortest peptides with two amino acids can act as powerful self-assembly motifs[1,2], giving rise to stable material, creating opportunities for rational functional design of materials with semiconductivity[3], piezoelectric[4,5], and fluorescence properties[6,7]. To develop these functional materials, both L- and D- amino acids, as well as unnatural amino acids, can be used. However, for in vivo biological applications, eukaryotic L-amino acid-containing peptides may be preferable as peptides with these amino acids are inherently biodegradable, making them more biocompatible and may be less toxic.

It is well-known that peptide self-assembly can form a variety of morphological structures, such as nanoparticles[8,9], nanofibers[10–12], nanotubes[1,13,14], and nanosheets[15,16]. The self-assembling process can be attributed from hydrogen bonding, salt bridges, metal chelation, hydrophobic interactions, π–π interactions, and/or van der Waals interactions[17] between the peptide chains. Although there are simple rules for the design of self-assembling peptides, based on known hydrophobic interactions, salt-bridges, and hydrogen bonding between peptide chains in water[18], the rules are empiric and far from meeting the application demands in various fields.

There are three common approaches for the development of self-assembling peptides. (1) Empirical design: most sequence design based on natural assembly peptides from biological systems such as KLVFF derived from parts of amyloid protein Aβ[16–20][19], and NFGAIL derived from a fragment of human islet amyloid polypeptide[20]. (2) Computational screening: the aqueous self-assembly propensity of tripeptides and dipeptides, such as PFF, KYF, and KFD were successfully screened and identified by computational method[21]. (3) Continuous enzymatic condensation induced hydrolysis and sequence exchange based on unprotected homo- and hetero-dipeptides (based on amino acid F, L, W, S, and D), creating a dynamic combinatorial peptide library for the discovery of self-assembling structures with different amino acid sequences and consequent nanoscale morphologies[22]. Nonetheless, there is a great need to create a more facile and unbiased tool for the rapid discovery of new self-assembling peptides.

One-bead one-compound (OBOC) combinatorial peptide library method, first described by Lam et al. in 1991[23], has been used extensively in the discovery of ligands against cell surface receptors, target proteins, host molecules for small molecules, protease substrates, kinase substrates, membrane-active peptides, small molecule inhibitors against galectin-1. OBOC library is not limited to natural amino acids; it offers a lot more structural possibilities, e.g., linear, cyclic, branch, and macrocyclic peptide libraries, as well as peptide libraries comprised of both natural and unnatural amino acids (L-/D-, α-/β-/γ-amino acids) and amino acids with post translational modifications such as phosphorylation, glycosylation, methylation, and glycation[24]. It can also be applied to the generation of small molecules, glycopeptides, lipopeptides, peptoids, and unnatural foldamers.

We believed the versatile and enabling OBOC combinatorial technology would enable us to develop self-assembling peptides efficiently if we could develop a simple but robust screening method to identify the self-assembling peptides on beads. We rationalized that fluorescent probes responsive to the hydrophobic environment will allow us to illuminate hydrophobic pockets formed by self-assembling peptides on the beads. One such fluorescent probe is nitro-1,2,3-benzoxadiazole (NBD),

which is known to fluorescent-activate under a hydrophobic environment[25]. NBD could be added as free dye or tethered to the peptide chain. We elected the latter for our peptide library design (Fig. 1).

## Results and discussion

**Fluorescent-activation screening assay.** It is well known that Aβ[16–20] peptide sequence KLVFF can self-assemble into nanofibers in water[19]. We, therefore, chose it as a model peptide to test our screening strategy and used KAAGG as the non-assembling negative control[26]. These control peptides, N-terminally capped with NBD were synthesized with standard Fmoc-chemistry, and TentaGel S resin beads as solid support but without any cleavable linker. After side-chain deprotection, the beads were thoroughly washed with DMF, which was replaced stepwise with water or methanol and then examined under a fluorescent microscope. In methanol, both (NBD)-KLVFF-beads and (NBD)-KAAGG-beads showed bright fluorescence. This is expected as the hydrophobicity-sensitive fluorescence dye NBD would fluoresce brightly under hydrophobic condition, in this case, methanol. In water, we expect fluorescent signal emitted from non-sequestered NBD to diminish, unless it is buried inside a hydrophobic environment such as within peptide aggregates or β-sheets formed by KLVFF peptides. This was exactly what was observed: under aqueous condition, (NBD)-KLVFF-beads fluoresced brightly but (NBD)-KAAGG-beads did not (Fig. 2a and Supplementary Fig. 1). Similar to KLVFF, two other peptides (NFGAIL, LIVGD)[27] with known self-assembling properties were found to yield similar results, and therefore were utilized as two additional positive controls (Supplementary Fig. 2). These results were gratifying as they demonstrated that such a simple assay would allow one to rapidly screen and identify self-assembling peptides from OBOC combinatorial libraries.

**Screening of new self-assembling peptides.** Once the screening method has been validated with our model KLVFF peptide, a totally random pentapeptide library with 19[5] permutations (cysteine was omitted from the library) was constructed using our previously published methods (Supplementary Fig. 3)[23]. Again, TentaGel S resin (loading of 0.31 mmol/g) was used as the solid

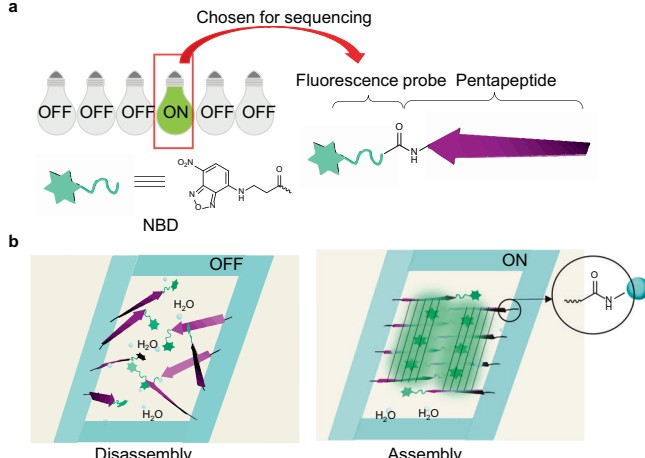

**Fig. 1 The principle for screening self-assembling peptides based on OBOC combinatorial peptide library. a** The screening process of self-assembling peptides by picking up the turn-on bead. The red line suggests the turn-on "bead". **b** Self-assembling peptides form hydrophobic pockets for fluorescent activation of N-terminally tethered organic dye nitro-1,2,3-benzoxadiazole (NBD). The blue band represents the bead matrix of crosslinked polystyrene. The black line indicates the linker chemistry.

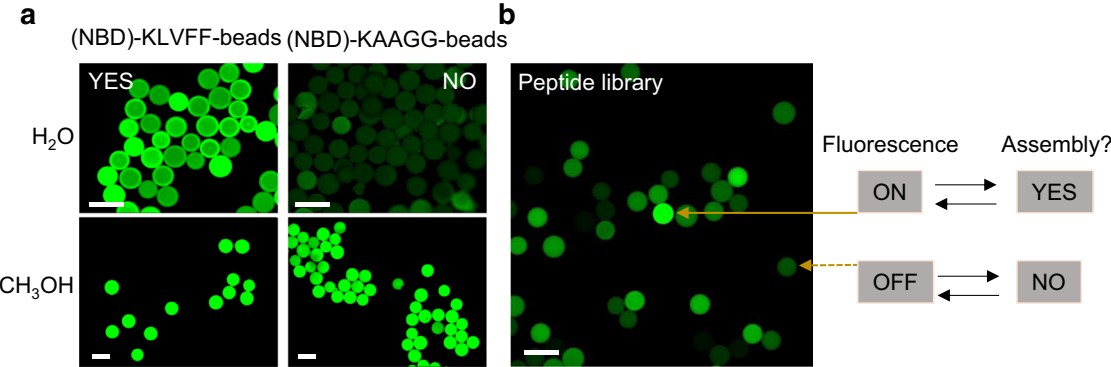

**Fig. 2 Screening assay development and OBOC library screening for self-assembling peptides. a** The fluorescence microscope imaging of positive control (NBD)-KLVFF-beads and negative control (NBD)-KAAGG-beads in water and methanol after incubation for 20 min. The scale bar is 200 μm. **b** Screening result of a representative region of a plate containing the OBOC library. The positive bead with an intense fluorescent signal was detected (pointed with solid arrow), the bead with diminished fluorescence was considered a negative bead (pointed with dotted arrow). The scale bar is 200 μm.

### Table 1 Peptide information screened from OBOC library.

| Penta-peptide | Fluorescence (beads) | Assembly | CMC (μM) | Charge[a] | GRAVY[b] |
|---|---|---|---|---|---|
| FTISD | On | Yes | 10.04 | − | 0.46 |
| ITSVV | On | Yes | 14.8 | 0 | 2.28 |
| YFTEF | On | Yes | 11.4 | − | 0.02 |
| ISDNL | On | Yes | 9.4 | − | 0.1 |
| LDFPI | On | Yes | 8.4 | − | 1.2 |
| FAGFT | On | Yes | 13.9 | 0 | 1.26 |
| FGFDP | On | Yes | 14.8 | − | 0.02 |
| FFVDF | On | Yes | 14.4 | − | 1.82 |
| RITWI | Off | No | | + | 0.58 |
| PLVKA | Off | No | | + | 0.86 |
| PFTTR | Off | No | | + | −0.94 |
| QIMRW | Off | No | | + | −0.5 |

[a]The charges for peptides were calculated according to the charges of amino acids.
[b]The grand average of hydropathicity (GRAVY) for each peptide was obtained using the ProtParam program (available at https://web.expasy.org/protparam/).

support. The pentapeptides in the library were N-terminally capped with NBD. Fig. 2b shows the screening result of a representative region of a plate containing the OBOC library. The positive bead with an intense fluorescent signal (pointed with a solid arrow) was noted. The rest of the beads were considered negative or weak positive beads. From a total of approximately 100 000 beads screened, we selected a total of eight strongest fluorescent positive beads and four dark beads as negative beads. These 12 beads were physically isolated for automatic Edman sequencing, and the result is shown in Table 1. The peptides were then resynthesized in soluble form and purified by high-performance liquid chromatography (HPLC) for testing (Supplementary Figs. 4–12). The amino acid sequences of the eight self-assembling pentapeptides are FTISD, ITSVV, YFTEF, ISDNL, LDFPI, FAGFT, FGFDP, and FFVDF. Not unexpectedly, of the 40 amino acids in these eight pentapeptides, 50% are hydrophobic residues, predominated by Phe (11 Phe, 4 Ile, 3 Val, and 2 Leu). For the remaining 20 amino acids, six are acidic residues (5 Asp and 1 Glu) and seven are neutral and hydrophilic (4 Thr and 3 Ser). The rest are 2 Pro, 2 Gly, 1 Tyr, 1 Asn, and 1 Ala. All eight peptides have at least two hydrophobic residues. Six out of the eight peptides have one negatively charged residue (Asp or Glu). The remaining two peptides are neutral. Interestingly even though our model peptide KLVFF has a Lys, none of the

eight identified self-assembling peptides has any basic residue. For the Pro containing pentapeptides, LDFPI and FGFDP, the Pro residue, which promotes peptide turns, resides at the fourth and fifth positions, but not in the middle. Not unexpectedly, all the four randomly selected negative non-assembling peptides (RITTR, PLVKA, PFTTR, and QIMRW) have very different sequences, and contain at least one positive charged amino acid.

To estimate the propensity of the identified peptides to self-assemble, the critical micelle concentrations (CMC) of the eight positive peptides and four negative peptides were determined, and the results are shown in Table 1 and Supplementary Figs. 13, 14. The CMC of all the eight positive peptides were detectable and found to range between 8.4 and 14.8 μM, indicating the propensity for these peptides to self-assemble under aqueous environment. In contrast, none of the four randomly selected negative peptides had a detectable CMC (Supplementary Fig. 14). Hydrophobicity of peptides can be predicted by the value of GRAVY[28] with more negative value for more hydrophilic peptides. The GRAVY values of both the positive and negative assembling peptides are summarized in Table 1. Of the four negative assembling peptides, two have a negative GRAVY value. For the rest of the peptides, both positive and negative assembling have GRAVY values ranging from 0.02 to 2.28, indicating that hydrophobicity alone is not a good predictor of self-assembling property. Experimental testing, such as the screening strategy described in this work, is needed. This is because multiple factors, including hydrophobicity, hydrogen-bonding, electrostatic interactions, and positions of each residue within the peptides, all contribute to the assembling and final morphology of the self-assembled nano-structure.

To estimate the number of self-assembling peptides in a random OBOC pentapeptide library, we first immobilized approximately 200 000 beads on the inside planar surface of a polystyrene dish using a solvent mixture of dichloromethane (DCM) and N,N′-dimethylformamide (DMF) at a ratio of 1:4 (v/v) as previously reported[29]. We then submersed the immobilized library beads in water. After equilibration for over 30 min, the entire plate was then scanned with a ZEISS LSM 800 confocal microscope (Carl Zeiss Microscopy, Thornwood, NY, USA) using an automatic stage. The stitched fluorescent images were then analyzed with MATLAB software to generate a fluorescent profile of the entire immobilized bead library, with relative fluorescent intensity ranging between 0 and 1.0 (Supplementary Fig. 15). Of the 186 288 beads screened, 79 or ~0.04% displayed a relative fluorescence intensity ≥0.99, and 559 or 0.3% beads displayed a fluorescence intensity ≥0.9. Peptides displayed on these positive beads are expected to have high propensity for self-assembling under aqueous condition.

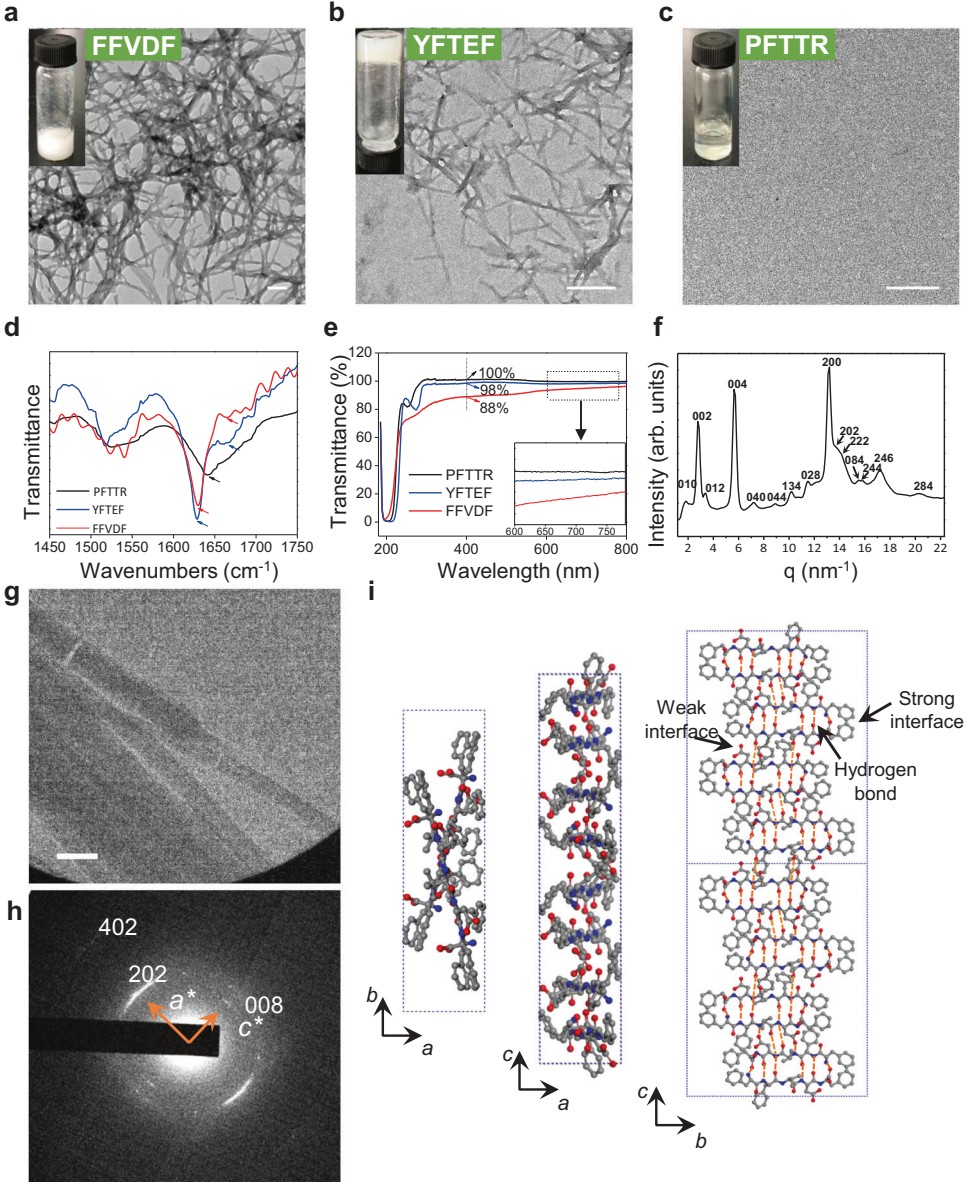

**Fig. 3 Structural characterization of identified self-assembling pentapeptides. a–c** TEM images of FFVDF, YFTEF, and PFTTR pentapeptides (30 mM in $H_2O$). Insets: photographs of the suspension, gel, and solution. The scale bar is 200 nm. **d** FT-IR absorption spectra in the amide I region of pentapeptides FFVDF, YFTEF, and PFTTR (30 mM in $H_2O$); narrowing and red-shifting of the amide modes of FFVDF and YFTEF indicate the presence of well-ordered assemblies for these peptides. **e** The transmittance spectra of FFVDF, YFTEF, and PFTTR at a concentration of 20 μM. **f** X-ray diffraction powder pattern of FFVDF nanofibrils. **g** Bright-field TEM image of nanofibrils self-assembled from FFVDF (30 μM in $H_2O$ with 1% DMSO). The scale bar is 200 nm. **h** Selected-area electron diffraction (SAED) pattern of the nanofibers corresponding to the part of BF image in (**g**). **i** *ab-, ac-, bc*-plane projections of the proposed molecular packing model of the nanofibers based on SAED and powder X-ray diffraction results.

**Evaluation of assembling structures of pentapeptides**. To further characterize these self-assembling peptides, we selected two positive peptides (FFVDF and YFTEF) and a negative control peptide (PFTTR) for the following studies: Fourier transform infrared (FT-IR) spectroscopy, UV−Vis spectra, and transmission electron microscopy (TEM). At 30 mM concentration in water, FFVDF generated a milky white solution, and TEM studies revealed an entangled fibrous network (Fig. 3a). YFTEF, dissolved in water at 30 mM was found to form a milky hydrogel, and an entangled fibrous network was observed under TEM (Fig. 3b). PFTTR (negative control) dissolved in water at 30 mM, as expected, showed a completely clear solution with no nanostructures observed under TEM (Fig. 3c). The results indicated that peptide FFVDF and YFTEF can self-assemble, and suggested

that the screening method employed here was reliable for the identification of self-assembling peptides.

Self-assembling peptide amphiphiles are often evaluated with FT-IR spectroscopy through characterization of β-sheets structure, with characteristic "amide I" infrared (IR) absorption bands, which are commonly found in these peptide amphiphiles[30]. For example, due to intermolecular hydrogen bonding involving the amide groups, the 1640−1655 $cm^{-1}$ amide I absorption band of free peptides in solution typically narrows and red-shifts to a lower frequency upon self-assembly. This phenomenon was observed in both FFVDF and YFTEF peptides (Fig. 3d), with the amide I absorption significantly narrowed and shifted to 1630 $cm^{-1}$, corresponding to a β-sheet-like arrangement of the amide groups, compared to the negative non-aggregating peptide PFTTR with

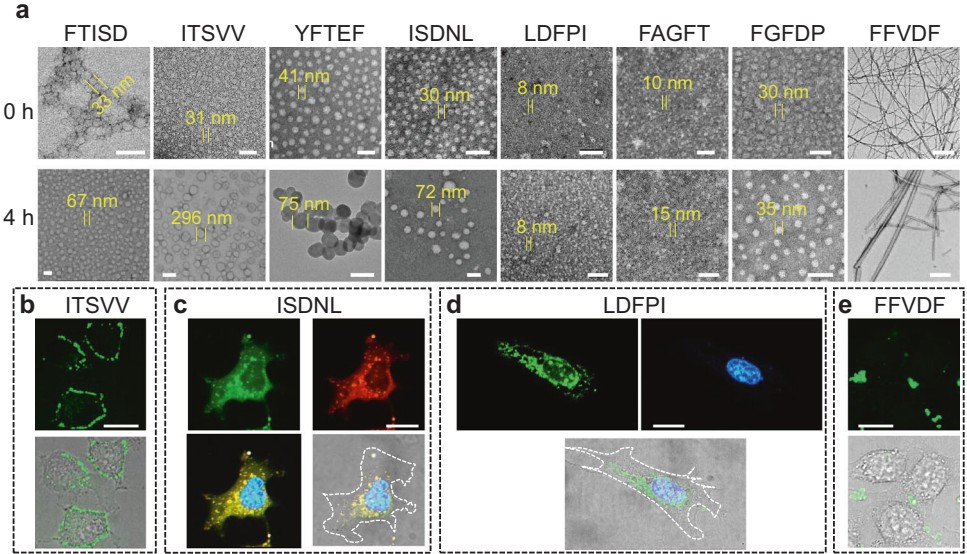

**Fig. 4 Structure transformation and cellular distribution of pentapeptide-based nanomaterials. a** TEM images of 40 µM chemically synthesized FTISD, ITSVV, YFTEF, ISDNL, LDFPI, FAGFT, FGFDP, and FFVDF "solubilized" in H$_2$O with 1% DMSO for 0 and 4 h. The scale bar is 100 nm for FTISD, YFTEF, ISDNL, LDFPI, FAGFT, and FGFDP at both 0 h and 4 h, and ITSVV at 0 h. The scale bar is 500 nm for FFVDF at both 0 and 4 h and ITSVV at 4 h. **b–e** Pentapeptide assemblies (40 µM) interaction with live HeLa cells. ITSVV (**b**) localized on the cell membrane, ISDNL (**c**) distributed inside the lysosomes throughout the cytoplasm, LDFPI (**d**) localized to the nucleus and perinuclear areas, and FFVDF (**e**) formed aggregates outside the cell. Green denotes fluorescein isothiocyanate (FITC)-labeled pentapeptides. Red (Lyso-Tracker Red) denotes lysosomes, blue (Hoechst 33324) is nucleus. The scale bar is 20 µm. The white dash line outlines the cell membrane.

absorption at 1641 cm$^{-1}$, signifying random coil structure. The absorption peaks at 1630 and 1664 cm$^{-1}$ of YFTEF and FFVDF are consistent with anti-parallel β-sheet structures.

To further confirm the peptide assembling in water, the solutions of pentapeptide were diluted to 20 µM for turbidity test. As shown in Fig. 3e, among the three peptide solutions tested, FFVDF showed the lowest transmittance in the range of visible light (400–780 nm). The transmittance of FFVDF, YFTEF, and PFTTR at 400 nm was determined to be 88, 98, and 100%, respectively, suggesting that FFVDF and YFTEF did self-assemble in water.

Given that FFVDF exhibited a strong propensity to self-assemble in water, it was further characterized with X-ray power diffraction experiment (Fig. 3f), TEM (Fig. 3g), and selected-area electron diffraction (SAED) technique (Fig. 3h). The results showed an ordered structure of FFVDF nanofibrils. The observed and calculated spacings of FFVDF nanofibrils were listed in Supplementary Table 1. According to the X-ray and electron diffraction data of the nanofibers (Fig. 3f, h), we deduced that the crystal structure of FFVDF in the nanofibers is orthorhombic with cell parameters of $a = 9.5$, $b = 35.3$, and $c = 45.6$ Å. Based on the above data, we established a packing model with one-unit cell, comprised of four dimers by adopting Cerius$^2$ modeling package. These four dimers oriented at two crossed directions as displayed in Fig. 3i. Each dimer is made up of two strands connected by hydrogen bonding. The hydrogen-bonding direction is along the $c$-axis. Strong or weak interface was formed within one dimer or between the neighboring two dimers, respectively, confirming the anti-parallel β-sheet structures.

**The interactions between self-assembling peptides and cells.** Self-assembling peptides with good biocompatibility may be used to construct nanocarriers for drug delivery. We recently reported the use of FFVLK, the reverse sequence of KLVFF, as a self-aggregating β-sheet forming domain for the delivery of HER2 targeting peptides for successful treatment of breast cancer in xenograft models[31]. Such FFVLK containing peptide is able to self-assemble into nanoparticles. Upon intravenous administration, the FFVLK-

nanoparticle was able to transform into nanofibrillar network when encountering HER2 at the tumor sites. To investigate the morphology of the nano-assembly formed by the eight self-assembling peptides, we dissolved each of these peptides in DMSO (5 mM) and then diluted them 100X with water at a final peptide concentration of 50 µM to induce spontaneous self-assembly. Each sample was fixed immediately after peptide solution preparation and at 4 h time point for subsequent TEM processing and analysis. As clearly shown in Fig. 4a, all eight self-assembling peptides initially formed some nanostructures: nanoparticles for ITSVV, YFTEF, ISDNL, LDFPI, FAGFT, FGFDP, and FTISD; nanofibers for FFVDF. Some of the nanoparticles were found to evolve over the 4 h time period to a larger size including FTISD, ITSVV, YFTEF, ISDNL, FAGFT, and FGFDP. In addition, 31 nm ITSVV nanoparticles were found to enlarge 10-times to a size of ~300 nm after 4 h. LDFPI (8 nm) maintained the particulate morphology without changing of diameter in 4 h. FFVDF formed nanofibers at 0 and 4 h. After 24 h, all the eight pentapeptides were found to transform to fibrils (Supplementary Fig. 16). These fibrils were then subjected to thermal annealing at 90 ℃ for 5 h, after which no significant change in morphology was observed under TEM (Supplementary Fig. 17), indicating that these peptide fibrils were at the thermodynamic minimum state. Time-dependent size variations of pentapeptides in water were further confirmed by DLS spectra (Supplementary Fig. 18). The initial diameters of nanoparticles formed by seven pentapeptides were larger than that observed by TEM, probably due to hydration radius. The size of the nanostructure increased over time, correlating well with the morphologic transformation. In contrast to the self-assembling peptides, no nanostructures were detected in any of the negative control PFTTR, RITWI, PLVKA, and QIMRW peptide preparations (Supplementary Fig. 19), which is consistent with the CMC results. Circular dichroism (CD) was used to monitor the secondary structures of the nano-assembly of the eight self-assembling peptides over time (Supplementary Fig. 20). Seven of the eight pentapeptides, except for FFVDF, showed stable CD spectra over 72 h, indicating no changes in secondary structures during the morphological

transformation. FFVDF, on the other hand, exhibited weaker cotton effect peaks in 72 h, indicating a decrease of chirality in FFVDF assemblies.

Finally, we examined the biological effects and cellular distribution of the eight self-assembling peptides on HeLa cells. CCK-8 cell viability assay results suggested that eight self-assembling peptides were non-toxic to HeLa cells up to a concentration of 200 μM (Supplementary Fig. 21). In order to observe the nanomaterials under fluorescent microscopy, each peptide preparation was spiked with 3% of the same peptide N-terminally labeled with FITC, and then added to HeLa cells at a final concentration of 40 μM. Our assumption was that FITC decorating 3% of the peptides would not significantly affect the nanostructure, cell uptake, and intracellular distribution. After 4 h of incubation with the peptide assemblies, the cells were observed under a fluorescent microscope (Figs. 4c, d and Supplementary 22). Obvious cellular uptake was observed in five of the eight peptides (YFTEF, ISDNL, LDFPI, FAGFT, and FGFDP). It appears that LDFPI nanoparticles might be able to permeate the cells and eventually be localized to the nucleus and perinuclear areas (Fig. 4d). It is important to note that LDFPI nanoparticle was small (8 nm diameter) even after 4 h incubation. Its small size may in part explain its ability to pass through the nuclear pore complexes[32,33]. YFTEF nanoparticles (41 nm evolved to 75 nm after 4 h) were found to be distributed throughout the cytoplasm but not the nucleus; both granular and diffuse staining were observed (Supplementary Fig. 22). ISDNL and FGFDP nanoparticles, both around 30 nm, appeared to distribute only to the cytoplasm, in granular forms. For ISDNL, the FITC green signal was found to co-localize with the Lyso-Tracker red signal, indicating that the nanoparticles were inside the lysosomes. FAGFT nanoparticle was small (10 nm evolved to 15 nm after 4 h) and appeared to mainly confine to the cytoplasm as well (Supplementary Fig. 22). FFVDF formed nanofibers at the outset and formed some fluorescent aggregates outside the cells (Fig. 4e). ITSVV, forming ~30-nm nanoparticles at 0 h, with 10-times size increase in 4 h, resulted in staining of the cell membrane and minor staining of the cytoplasm (Fig. 4b and Supplementary Fig. 22). FTISD, with size increase over time, stained both the cell membrane and cytoplasm (Supplementary Fig. 22). To evaluate the cell uptake mechanism of peptide assemblies (YFTEF, ISDNL, and FGFDP) into the cytoplasm, we treated the cells at 4 °C or with various endocytosis inhibitors (Supplementary Fig. 23). Based on the result of this study, we have determined that endocytic uptake of YFTEF was clathrin-dependent; uptake of ISDNL was through caveolae-dependent endocytosis and micropinocytosis; FGFDP uptake was both caveolae-dependent and clathrin-dependent. Although all three peptide assemblies (YFTEF, ISDNL, and FGFDP) ended up in lysosome (Fig. 4c and Supplementary Fig. 24), their intracellular trafficking kinetics were found to be different. Through time-dependent fluorescence co-localization studies with dye-staining lysosome and FITC-labeled peptides (Supplementary Fig. 24), we have found that YFTEF remained in lysosomes for 8 h without obvious escape. In contrast, both ISDNL and FGFDP were found to escape from lysosomes in approximately 4 and 1.5 h, respectively.

Here, we show the development of a unique but very simple fluorescent-activation screening method, which has allowed us to rapidly discover self-assembling short peptides for nanomaterial development. NBD placed at the N-terminus of a random penta-peptide library will fluoresce when the displayed peptides self-assemble to form hydrophobic pockets that interact with NBD; free dyes may be also used for screening self-assembly peptides (Supplementary Fig. 25). This proof-of-concept report validates that the method works well and provides a tool, not only to discover biological useful nanomaterial, but also allows us to

identify peptide motifs for self-assembling, in a non-bias manner. As increasing number of peptides are identified with this approach, we may be able to better understand how peptides self-assemble. Although the focus of this report is on self-assembling of the same peptide, we can easily design OBOC libraries for the discovery of hybrid nanomaterial formed by self-assembly of two or more different peptides. Because OBOC libraries are produced by chemical synthesis, the discovered self-assembled nanomaterials are not limited to canonical amino acids. Various nanomaterials with a huge range of building blocks can be developed. The screening process is very simple and can be easily automated. Physicochemical conditions such as pH, ionic strength, solvents, temperature, electric current, and magnetic field can also be incorporated into the screening steps. Sequential screening of immobilized OBOC library-beads[29] under various conditions, in conjunction with optical spectral analysis, will enable us to discover nanomaterial with desirable properties, including stimuli-responsive property that is very important in biomedical applications.

All eight positive pentapeptides identified in this study were found to be non-toxic to HeLa cells up to at least 200 μM concentration, and have the capacity to self-assemble to form either nanoparticles or nanofibers. Several of these peptidic nanostructures were found to interact with living mammalian cells, either through cell surface binding, or enter cells into the lysosomes, cytoplasm, or inside the nucleus. Work is currently underway to use some of these peptides for drug and nucleic acid delivery. As more self-assembling peptides with various unique properties are discovered, we may be able to combine them to make nanomaterials for various biomedical applications.

## Methods

**Materials**. TentaGel S Resin was purchased from Rapp Polymere (Germany, loading 0.31 mmol/g). 9-Fluorenylmethoxycarbonyl (Fmoc)-protected amino acids, 2-(1H-benzotriazole-1-yl)-1,1,3,3- tetramethyluronium hexafluoro-phosphate (HBTU), and Wang resin were obtained from GL Biochem (China). Trifluoroacetic acid (TFA), fluorescein isothiocyanate (FITC), N-methyl morpholine (NMM), piperidine, and N,N′-dimethylformamide (DMF) were all from Beijing Chemical Plant (China). 1,2-Ethanedithiol (EDT) was purchased from Alfa Aesar (USA). Dimethyl sulfoxide (DMSO) was purchased from Aldrich Chemical Co. and used without further purification. Cyanogen bromide (CNBr) was from J&K Chemical (China). The HeLa cell lines were received from the Cell Culture Center of the Institute of Basic Medical Sciences, Chinese Academy of Medical Sciences (Beijing, China). Cell culture medium and fetal bovine serum were from WisentInc (Multicell, WisentInc, St. Bruno. The cell counting kit-8 assay (CCK-8) (Beyotime Institute of Biotechnology, China) was used. Hela cells were maintained Dulbecco's modified eagle's medium (DMEM) with 10% fetal bovine serum and 1% penicillin. All cells were cultured in a humidified atmosphere containing 5% $CO_2$ at 37 °C.

**The construction and synthesis of OBOC libraries with NBD**. The OBOC pentapeptide libraries (19[5]) were built using Fmoc strategy SPPS (solid-phase peptide synthesis) (Supplementary Fig. 3). TentaGel S Resin (named N-beads with loading 0.31 mmol/g) was used as the solid phase support. The pentapeptide sequence of $X_1 X_2 X_3 X_4 X_5$ was constructed in the library, in which $X_1–X_5$ represents either Ala, Asp, Glu, Phe, Gly, His, Ile, Lys, Leu, Met, Asn, Pro, Gln, Arg, Ser, Thr, Val, Trp, and Tyr residues. During the synthesis of OBOC library, solid support beads were mixed and split equally in each cycle, then different amino acids were added, separately[23], the process was repeated four times. The synthesis of peptides was used anhydrous DMF as solvent. The Fmoc group was removed by 20% (v/v) piperidine in DMF and the deprotection time was 10 min twice. During the coupling step, the HBTU (4 mM) and Fmoc-amino acid (4 mM) were dissolved in DMF containing NMM (0.4 mM). The coupling time was 50 min. Qualitative Fmoc deprotection and coupling were confirmed by ninhydrin test. All the above experiments were carried out in the solid phase peptide synthesis vessels with sieves in it. After the last Fmoc was removed, the beads were washed and dried over vacuum overnight, then the beads were swollen in water for 24 h, outer layer reacted with Fmoc-OSu and inner core reacted with (Boc)₂O to prepare bilayer beads[34]. After Fmoc-deprotection of the outer layer, NBD-COOH was conjugated to the N-terminus of the library peptide. The bilayer bead pentapeptide library was ready for screening for self-assembling peptide after deprotection with a cocktail of cleavage reagents (95%, TFA: 2.5% water: 2.5% EDT v/v).

**Synthesis of pentapeptides**. Positive and negative peptides by self-assembly screening process were de novo synthesized by solid-phase methods using standard Fmoc-Chemistry. Wang resin was used as the solid phase support. The synthesis process was basically consistent with the above SPPS peptide synthesis. After elongation, a cleavage cocktail (95%, TFA: 2.5% water: 2.5% EDT v/v) was used for cleaving the peptide from beads for 3 h. The TFA was removed by evaporating using vacuum rotary to obtain a concentrated product. Then the product was precipitated in cold anhydrous diethyl ether, centrifuged, dried to obtain the crude peptides. The peptides were purified by preparative reversed-phase high-performance liquid chromatography (HPLC) with a preparative reversed-phase Inertsil C18 HPLC column (ODS-3, 5 μm, 20 × 250 mm). A linear gradient of acetonitrile/water with 0.1% TFA respectively from 5/95 (v/v) to 70/30 (v/v) during 18 min, then 70/30 (v/v) to 90/10 (v/v) in 4 min and in this flow continue 1 min, next return to 5%/95% (v/v) tilled for 3 min was used as the mobile phase. The separation was performed with a flow rate of 1 mL/min and the monitoring wavelength was 220 nm using a UV detector (Waters 2535Q). The purified peptides were analyzed matrix-assisted laser desorption ionization time-of-flight mass spectrometry (MALDI-TOF-MS, Bruker Daltonics, USA) and liquid chromatography-mass spectrometry (LC-MS 8050, Shimadzu, Japan).

**Preparation of peptide-based nanomaterials**. Pentapeptides monomers were dissolved in DMSO at a concentration of 5.0 mM, followed by quick injection into water at a volume ratio of 1:99 for DMSO and $H_2O$ to obtain the nanomaterials solution (50 μM). For 30 mM pentapeptide solution, the pentapeptide was dissolved in water and sonicated for 3 min.

**TEM for the morphology**. The pentapeptide solution with the concentration of 20 μM (10 μL) was placed dropwise onto a copper mesh for 5 min, then most of the liquid was removed through a filter paper. Ten microliters of uranyl acetate solution were employed to stain the samples for 5 min, followed by drying the spare liquid with the filter. Finally, the copper mesh was washed with 10 μL of deionized water, which was blotted after staining and drying at room temperature. All of the samples were observed by TEM (Tecnai G2 20 S-TWIN) at an accelerating voltage of 200 kV.

**TEM and SAED**. We used JEOL-1400 to capture the BF image and SEAD patterns of FFVDF nanofibrils at 110 kV. A selected-area aperture was inserted in electron diffraction mode. A water droplet containing the nanofibrils was cast onto copper grids with the supporting film.

**FT-IR and XRD spectra**. The pentapeptide with a concentration of 20 μM freshly prepared which was used for FT-IR measurement. The solution mentioned above lyophilized to obtain the powders for XRD measurements. The FT-IR spectrum was recorded on a spectrometer (Spectrum One, Perkin Elmer Instruments Co., Ltd.). The XRD spectrum was recorded on a Xeuss SAXS/WAXS system (Xenocs Asia Pacific Pte., Ltd.).

**CD spectra**. The CD spectra of pentapeptides (40 μM) were collected at room temperature using a CD spectrometer (JASCO-1500, Tokyo, Japan) with a cell path length of 1 mm. The measurements were implemented between 190 and 230 nm with a resolution of 1.0 nm and a scanning speed of 300 nm/min. For each measurement, three spectra were collected and averaged.

**CLSM observation**. Hela cell line was purchased from cell culture center of Institute of Basic Medical Sciences, Chinese Academy of Medical Sciences (Beijing, China). Hela cells cultured with pentapeptide nanomaterials (50 μM) were imaged using a Zeiss LSM710 confocal laser scanning microscope (Jena, Germany). The Hela cells were seeded in complete DMEM in a humidified atmosphere with 5% $CO_2$ and then cultured at 37 °C overnight. Then, 1 mL of serum-free fresh medium containing pentapeptide (50 μM) was used for replacing the medium, and the cells were cultured for 4 h and washed with PBS three times before being imaged using a Zeiss LSM710 confocal laser scanning microscope with a 40× objective lens. The cell lines had been authenticated utilizing short tandem repeat DNA profiling. All cells were tested negative for cross-contamination of other human cells and mycoplasma contamination.

**Statistical analysis**. All data are reported as the mean ± standard deviation (s.d.). The in vitro experiments were performed in three independent experiments with at least three technical replicates.

**Reporting summary**. Further information on research design is available in the Nature Research Reporting Summary linked to this article.

## Data availability
The data that support the findings of this study are available from the corresponding author upon reasonable request.

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

## Acknowledgements

This work was supported by the National Natural Science Foundation of China (51890891, 51890894, 51725302, 21807020, 51573031, and 51573032), Science Fund for Creative Research Groups of the National Natural Science Foundation of China (11621505). The Combinatorial Chemistry and Chemical Biology Shared Resource at University of California Davis assisted the synthesis of OBOC library and sequencing decoding of beads, utilization of this Shared Resource was supported by the UC Davis Comprehensive Cancer Center Support Grant NCI P30CA093373. D.S. was supported by a training grant [GM 113770] from the National Institute of Health.

## Author contributions

L. W., R. L. and K. S. L. conceived the project. P. P. Y., Y. J. L., L. W., H. W., R. L. and K. S. L. planned and designed the experiments. P. P. Y., Y. J. L., L. W. and R. L. performed most of the experiments. Y. C. and Z. W. L. performed the structural analysis. L. Z. assisted with the chemical synthesis, and K. Z., W. W. X. and H. C. assisted with the cell culture studies. J. Q. W. and D. S. assisted the experiments in revision. P. P. Y., Y. J. L. and Y. C. contributed equally to this work.

## Competing interests

The authors filed patents pertaining to the results presented in the paper. The authors declare the following competing financial interest(s): L. W., H. W., P. P. Y. and Y. J .L. are the co-inventors of a pending patent. The remaining authors declare no competing interests.
