## [Peer Review File · Nature Communications]

The previous round of reviews came from another journal. The correspondence below started after the paper was transferred to Nature Communications.

Author's response to the reviewer #1's comments:

Reviewer #1: This paper describes an interesting approach to the discovery of self-assembling peptide motifs by combinatorial searching of the pentapeptide sequence space. The approach is based on one-bead-one-compound combinatorial screening where peptides are directly synthesized on water swelling tentagel beads and screened for self-assembly propensity through functionalization with a self-assembly-reporting fluorophore. To my knowledge, this is a new method that could in principle reveal interesting design features for short peptides by rapidly screening large sequence sets followed by evaluation of the selected peptides in aqueous self-assembly.

The approach has a number of strengths but the paper currently does not deliver on its promise, in that the authors opted to analyze only a very small subset of the sequence space (12 out of 2.5 million). The eight self-assembling candidates have some interesting features (typically containing 2 aromatics), but the approach does not give rise to new design rules or insights. I am enthusiastic about the approach, but the authors are encouraged to consider the following suggestions to be able to demonstrate better what this methodology offers as a peptide design approach. It is not the objective to gain insights on the entire population of 2.5 million peptides, but to screen and pinpoint the self-assembling peptides. How many of the 100,000 beads that were analyzed are considered 'positive'. Is it possible to use e.g. FACS to quantify the fluorescence of a large population of beads, and then show a distribution, and specifically indicate where the selected 8 exist within this population? It is currently not known what percentage of the pentapeptide space is prone to self-assembly- the authors have an opportunity to address this question.

Response:

In response to the reviewer's suggestion, we performed further experiment to quantify the fluorescence of 536 beads randomly sampled from an OBOC pentapeptide library (Figure R1). Most of these beads (> 95 %) were "fluorescence on" when placed in DMF and H₂O (9:1). The solvent was then replaced with H₂O. After 2 h of incubation, only 15 beads or approximately 2.7% of the beads were found to remain highly fluorescent, and they are expected to be self-assembling peptides.

Figure R1. A small OBOC pentapeptide library of 536 beads was examined under fluorescent microscopy. Most beads fluoresced when incubated with DMF/H₂O (9:1) (left panel), but only a fraction of these beads remained strongly fluoresce when the solvent was replaced with H₂O (right panel). Those are positive peptides with high self-aggregation propensity.

2. *What is the role of the nitro-1,2,3-benzoxadiazole (NBD) fluorophore and linker in the self-assembly process? It seems reasonable that the N-terminal modification will enhance the self-assembly propensity of these peptides. Please comment.*

Response:

Nitro-1,2,3-benzoxadiazole (NBD) fluorophore will “turn on” (fluoresce) when placed in a hydrophobic environment, but “turn off” in a hydrophilic environment. Therefore, NBD was selected as a convenient probe to detect hydrophobic pockets or crevices formed by peptide self-assembling. NBD could be tethered to the N-terminus or remain free in solution during screening. In our initial screening experiments, we linked NBD to the N-terminus of the peptides displayed on the outer layer of the topologically segregated bi-layer bead (see below).

We cannot totally rule out that NBD would not have any enhancing effect on the self-assembling of some of the peptides. However, we can comfortably state that NBD placed at the N-terminus of the peptide will not diminish the validity of using OBOC library to screen for self-assembling peptides. This assertion is supported by: (1) All 8 identified peptides, when resynthesized without NBD could still self-assemble, (2) large proportion of the beads within the OBOC library, all N-capped with NBD, did not fluoresce under aqueous condition (Figure R1), and (3) NBD is water soluble.

The synthetic scheme of OBOC library, illustrating how NBD was incorporated into the outer layer of the beads is shown in Figure R2 and Figure S2.

Figure R2. Synthetic scheme for the preparation of OBOC peptide library N-capped with nitro-1,2,3-benzoxadiazole (NBD) at the outer layer of the bead for self-assembling peptide discovery.

3. What is the reason for opting for a covalently attached dye (NBD), rather than a non-covalent reporting dye that could diffuse into the beads- e.g. congo red or thioflavin T.

Response:

As indicated above, self-assembling peptide screening can be achieved with either placing NBD at the peptide N-terminus or adding NBD as free reporter dye. One distinct advantage of using N-tethered NBD is that an immobilized OBOC library can be screened sequentially under various conditions, e.g. ionic strength, pH, metal ions, and solvent, without the concern on the solubility of NBD under these various conditions; sequential screening of such library is much more convenient, without the need for prolonged equilibration after each condition. Furthermore there is concern on using free NBD as probe because some of the screening conditions may impair the solubility and therefore diffusion of NBD into the beads to turn the bead "on". That being said, we did confirm that free reporting dye such as NBD, Nile red, and thioflavin T (ThT) could be used as probe to screen for self-assembling peptides, as suggested by the reviewer. In this experiment, we used KLVFF as positive control peptide and KAAGG as negative control. As shown in Figure R3 and Figure S25 in supporting information, all these three dyes could turn KLVFF-beads "fluorescent on" under aqueous condition, but KAAGG-beads remained "fluorescent off" under the same condition.

Figure R3. Fluorescent microcopy of KLVFF-bead and KAAGG-bead after incubation with free ThT, Nile Red, or NBD. NBD generated much stronger signal than the other two dyes.

4. It is recommended to analyse more than one positive control to convincingly demonstrate that the approach works for a variety of sequence designs. NFGAIL or LIVGD (PNAS January 25, 2011 108 (4) 1361-1366) could be tested, for example.

Response:

In response to reviewer's suggestions, we carried out the suggested experiment by adding NFGAIL and LIVGD as positive controls. All three (NBD)-KLVFF-beads, (NBD)-NFGAIL-beads and (NBD)-LIVGD-beads show "fluorescence-on" when the beads were placed in water. These results further validate our screening strategy for self-assembling peptides. (Figure R4 and Figure S2).

Figure R4. Fluorescent micrograph of NFGAIL-, LIVGD-, KLVFF-, and KAAGG-beads N-capped with NBD. Limiting level (5% substitution) of NBD was used during the N-capping, resulting in preferential coupling

of NBD to the peptides on the outer shell of the beads. As a result, fluorescent activation was higher at the outer shell.

5. *The time dependent and dynamic nature of peptide assembly: whether these assemblies represent the thermodynamic minimum could be assessed further by thermal annealing.*

Response:

According to the reviewer's suggestion, we conducted thermal annealing experiments to study peptide assembling and stability of assembled nanostructures to heat. Assembled peptides were annealed at 90 °C for 5 h. The annealed samples were visualized by TEM. As shown in Figure R5 and Figure S16 in supporting information, all eight pentapeptides showed the morphology of nanofibers, which were very similar to the morphology of these pentapeptides in water for 24 h prior to heating (see Figure R7 below), indicating that these peptide assemblies at 24 h in water were at the thermodynamic minimum state.

Figure R5. The TEM images of pentapeptides FTISD, ITSVV, YFTEF, ISDNL, LDFPI, FAGFT, FGFDP and FFVDF, after thermal annealing at 90 °C for 5 h.

6. *The authors should comment on the dielectric constant and viscosity inside these beads; as this could impact on solvation, ionization propensity and consequent self-assembly behavior?*

Response:

We thank reviewer for the suggestion. Unfortunately, we do not have the instrument to measure the dielectric constant and viscosity inside the solid bead (~100 μm bead). We are hoping to collaborate with others in the future to systematically study how dielectric constant and viscosity inside the TentaGel beads affect peptide solvation, ionization propensity and consequent self-assembly behavior. This however will be subject of future reports.

7. *The full characterization of just two candidates seems minimal and does not really give new insights. It is recommended that the number of fully characterized peptides is expanded, for example to study particle versus fiber forming peptides, and look for sequence features related to this.*

Response:

According to the reviewer’s suggestion, we performed additional experiments to fully characterize the morphological states of all 8 pentapeptides. The time-dependent morphology of these peptides at 0, 4, and 24 h was shown in Figure R6-7 (Figure 4a in manuscript and Figure S15 in supporting information). Below is a discussion on the amino acid composition and sequence in relation to the self-assembling propensity. The three aromatic phenylalanines of FFVDF may be the driver to form highly-ordered fibrous structures. The proline residue in LDFPI may lead to disordered packing to form small nanoparticles.

Figure R6. TEM images of 40 μ M chemically synthesized FTISD, ITSVV, YFTEF, ISDNL, LDFPI, FAGFT, FGFDP and FFVDF pentapeptides “solubilized” in H₂O with 1% DMSO for 0 and 4 h. The scale bar is 100 nm for FTISD, ITSVV, YFTEF, ISDPI, LDFPI, FAGFT and FGFDP, and 500 nm for FFVDF.

Figure R7. The TEM images of the pentapeptides FTISD, ITSVV, YFTEF, ISDNL, LDFPI, FAGFT, FGFDP, and FFVDF in water with 1% DMSO after 24 h.

Minor:

- Use of an abbreviation in the title should be avoided.

Response:

We have modified the title “Rapid Discovery of Self-Assembling Peptides with One-Bead One-Compound Peptide Libraries”.

- Cartoon fig 1b what is the blue line? Please show the linker chemistry.

Response:

Figure 1b has been revised and the blue bands represent bead matrix of polystyrene to which the peptides were tethered, via a 2000 dalton linear polyethylene glycol linker. The linker used in NDB attachment to N-terminus is shown in revised Figure R8a and R8b.

Figure R8. The principle for screening self-assembled peptides based on one-bead one-compound (OBOC) combinatorial peptide library. Self-assembling peptide forms beta-sheet under aqueous condition, creating hydrophobic pockets for fluorescent activation of N-terminally tethered organic dye nitro-1,2,3-benzoxadiazole. The blue band represents the bead matrix of crosslinked polystyrene.

- Figure 2a- there seems to be core/shell differences in fluorescence? Please comment.

Response:

As shown in the synthetic scheme for the OBOC library (Figure R2), topologically segregated bi-layer beads was created with NBD N-capping the peptide on the outer layer and peptides at the inner core of the bead remain N-terminally free for Edman sequencing. As a result, only the shell fluoresced in the library beads.

For the positive control and negative control peptide beads, limiting level (5% substitution) of NBD was used during the N-capping, resulting in preferential coupling of NBD to the peptides on the outer shell of the beads. As a result, fluorescent activation was higher at the outer shell.

Author's response to the reviewer #2's comments:

Referee #2 (Remarks to the Author): This elegant work describes a unique way to select for short self-assembling peptides using one-bead one-compound peptide library technology. The idea is straightforward, yet the results are notably interesting and important. I really like the innovative application of the OBOC method for this search. The work is well related to the timely interest in peptide building blocks for nanotechnological and materials science applications. The new methodology could allow a significant increase in the chemical space of peptide modules used for such applications. I assume that it will now be followed-up by other groups for longer and shorter sequences. The manuscript is well-written and the conclusions are being supported by the experimental work. Here are a few comments to be considered:

1. The peptide sequences that were found are already long enough to be searched for in proteins? Is there a lower than expected representation of highly aggregative sequences? Or alternatively, is there use of the modules for cell penetration?

Response:

We thank reviewer's positive comments and suggestions. Since pentapeptide is relatively short, when we search the protein data base, we found protein sequence matching in all 8 identified peptides. However the significance of these peptide segments in protein folding or functions are not clear. Table R1 listed 10 proteins with -FTISD- sequence. Search results for the other 7 proteins are not shown.

As indicated in Figures 4 and S21-23, many of the 8 identified pentapeptides do have the capacity to self-assemble into nanoparticles and to enter cells through endocytosis and/or other mechanisms.

Table R1. The research results of FTISD in protein library.

No.	Protein	The position of FTISD in protein
1	Cytochrome c oxidase subunit 3	189 FTISD 193
2	Phosphatidylglycerol--prolipoprotein diacylglyceryl transferase	122 FTISD 126
3	Glucosamine-6-phosphate deaminase	246 FTISD 250
4	Cilia- and flagella-associated protein 300	138 FTISD 142
5	Protein LNK3	146 FTISD 150
6	L-lactate dehydrogenase	290 FTISD 294
7	Ribosomal large subunit pseudouridine synthase C	14 FTISD 18
8	Protein YIPF1 homolog	327 FTISD 331
9	Acyl-lipid (9+3)-(E)-desaturase	30 FTISD 34
10	Exodeoxyribonuclease 7 large subunit	240 FTISD 244

2. If indeed this is predictive, it might be that future hexapeptide or heptapeptide OBOC search will be useful. How feasible is it?

Response:

We agree with the reviewers that similar study can be employed in search for self-assembling hexapeptides and heptapeptides. This can be readily achieved.

3. The FFVDF peptide, which exhibited a strong propensity to assemble in water, is similar to the highly studied diphenylalanine peptide. In the case of the diphenylalanine, various morphologies could be achieved under different conditions. It is also the case with the pentapeptide?

Response:

We thank the reviewer's insightful suggestion. We found in the literature that dipeptide FF can form nanotube in 1,1,1,3,3,3 hexafluoro-2-propanol (Science, 300, 2003, 625-627) and nanofibers in water (Adv. Mater. 22, 2010, 583-587). Similarly, we studied the morphology of pentapeptide FFVDF under the same conditions and the results are shown in Figure R9. The assembly of FF was a little different from that of FFVDF in different solvents. They showed different morphologies.

Figure R9. TEM images of FF and FFVDF in different solvents a), d) in CH_2Cl_2 ; b), e) $\text{CH}_2\text{Cl}_2/\text{C}_2\text{H}_5\text{OH}$ with systems with ethanol contents of 5%; c), f) $\text{HPIF}/\text{H}_2\text{O}$ with HPIF contents of 2%. The FF results (a,b, and c) were obtained from the reference (ACS Nano 2016, 10, 2138-2143). The FFVDF results (d,e, and f) were obtained in our laboratory.

4. Also related to the diphenylalanine, how similar are the crystal structures of the dipeptide and pentapeptide.

Response:

The crystal structure of FF was obtained from the literature (Chem. Eur. J. 2001, 7, 5153-5159), and compared with our pentapeptide FFVDF. Both structures contain sheet-like configuration through hydrogen bonding. In dipeptide or pentapeptide, the most common structural motif contains two $\text{NH}_3^+\cdots\text{OOC}^-$ head-to-tail hydrogen bonds that rolled into two-dimensional sheet. The group of solvent molecules could accept the residual amino proton and form crystal structure including water or other solvent molecules. In FF, the crystal structure contains no solvent molecules and the two adjacent linear peptide backbones pack side by side that resemble into a parallel β -sheet-like configuration. Thus, it could form layer to layer structure.

5. How similar are the results in the computational study of tripeptides (reference 21) as compared to the current results with the pentapeptides?

Response:

According to the reviewer's suggestion, we compared the current results with that in the computational study of tripeptides reported in reference 21 (Nat. Chem. 7, 30-37 (2015)).

In our study, the amino acid sequences of the 8 self-assembling pentapeptides are FTISD, ITSVV, YFTEF, ISDNL, LDFPI, FAGFT, FGFDP and FFVDF. Not unexpectedly, of the 40 amino acids in these 8 pentapeptides, 50% are hydrophobic residues, predominated by Phe (11 Phe, 4 Ile, 3 Val, and 2 Leu). For the remaining 20 amino acids, 6 are acidic residues (5 Asp and 1 Glu) and 7 are neutral and hydrophilic (4 Thr and 3 Ser). The rest are: 2 Pro, 2 Gly, 1 Tyr, 1 Asn, and 1 Ala. All eight peptides have at least two hydrophobic residues. Six out of the 8 peptides have one negatively charged residue (Asp or Glu). The remaining two peptides are neutral. Interestingly even though our model peptide KLVFF has a Lys, none of the 8 identified peptides has any basic residue. For the Pro containing pentapeptides, LDFPI and FGFDP, the Pro residue which promote peptide turns, resides at the fourth and fifth positions, but not in the middle. Not unexpectedly, all the four randomly selected negative non-assembling peptides (RITTR, PLVKA, PFTTR, QIMRW) have very different sequences, and contain at least one positive charged amino acid.

For the tripeptides, reference 21 mentioned that "aromatic amino acids (F, Y and to a lesser extent W) are more favorable in position 2 and 3 (from N-terminus to C-terminus is position 1, 2 and 3); negatively charged amino acids (E and especially D) are strongly favored in position 3, and positively charged and hydrogen-bond-donating amino acids (K, R; S, T) promote self-assembling when located at the N-terminus.

Also, proline is favored in position 1, which could be because of its unique conformational properties allowing better packing of the short peptides”.

Reference 21 shows that many self-assembling tripeptides with high aggregation score (AP) have hydrophobic, negatively charge and positively charged residues. In our study, besides hydrophobic residues and the negatively charge residues, none of our 8 pentapeptides has positively charge residues, indicating that the self-assembling rules are somewhat different in tripeptides and pentapeptides..

From both studies, we can conclude that (1) hydrophobic and negatively charged residues, such as F, Y, D, and E are important for self-aggregation propensity, and (2) P is favored at end of the peptide due to the unique conformational properties. The fact that no positively charge residues are seen in any of our 8 peptides may reflect that longer pentapeptide might not rely so much on salt-bridges for self-aggregation as the shorter tripeptide might. It is still too early to draw any definitive conclusion. We envision that the method reported in this manuscript will allow us to generate large volume of self-aggregation propensity data on 2-mer, 3-mer, 4-mer, 5-mer, 6-mer, and 7-mer peptides in an unbiased manner. When analyzed in conjunction with computational modeling, one may be able to begin to understand peptide assembling.

Author’s response to the reviewer #3’s comments:

Referee #3 (Remarks to the Author):

Summary of the key results:

The authors screened an OBOC library for peptide sequences that aggregate in the aqueous solution and identified eight new peptide sequences that aggregate into different forms of nano structures after resynthesis and purification. Some of the peptides form nanofibers while others enter the mammalian cell to different extents. The latter is potentially very interesting/useful, if properly established.

Originality and significance:

This represents a very novel application of the OBOC library, in this case, the discovery of potential new materials. In fact, the finding is somewhat surprising, as resin-bound molecules usually behave differently from those in solution. The significance is less clear, for the reasons described below.

Response:

We have performed several additional experiments including cell uptake mechanistic studies. We hope the significance of this work is becoming more apparent.

Data & methodology:

The work is relatively preliminary. While the methodology may be very useful in the discovery of new materials, this potential has not yet been clearly demonstrated. A key question is whether the authors can control the type and/or properties of the materials derived from screening such libraries. If yes, the method would be very powerful; if not, the utility of the approach would depend on whether the authors can design the proper assay for a particular type of structure. The latter seems to be challenging. In this study, the authors found that some of the peptides (as nanoparticles) have the capacity to enter HeLa cells. This is potentially very interesting, but the data are still preliminary.

Key questions include: 1) Do they get into the cytosol? and 2) how efficiently do they get into cell/cytosol? Confocal microscopy is a good first step for answering these questions, but it is not quantitative or definitive. More definitive (e.g., functional) assays for cytosolic entry and comparison with proper benchmarks will be necessary, as many nanoparticles have already been demonstrated to enter the mammalian cell by endocytosis, but are usually entrapped inside the endosome and lysosome. Mere delivery of nanoparticles into the endolysosomal compartments is not practically useful.

Response:

The power of OBOC technology is that many unbiased huge random peptides can be created easily and economically, and the parallel bead-based screening approach affords a large number of different assays, ranging from protein binding assay, to cell-based functional assay, to chemical reaction assays, to biophysical assays, to material property assays. Positive beads identified can then be decoded and resynthesized in larger quantity for full characterization, on bead or in solution. Not all positive beads identified in the primary screen are expected to be true positive, or be endowed with a precise desirable property. Full characterization of the identified hits will allow us to select the hit with desirable properties. This has been our approach for our published work on many biological targets.

In this work, we aim to discover self-assembling peptides. It is gratifying to see that in our first screening experiment, all 8 identified peptides have self-assembling properties. Many of them form nanoparticles, some of them form nanofibers during the initial phase of self-assembling. In the conclusion section of our manuscript, we wrote: "This proof of concept report validates that the method works well and provides a new tool, not only to discover biological useful nanomaterial, but also allows us to identify novel peptide motifs for self-assembling, in a non-bias manner. As increasing number of peptides are identified with this approach, we may be able to better understand how peptides self-assemble. Although the focus of this report is on self-assembling of the same peptide, we can easily design OBOC libraries for the discovery of hybrid nanomaterial formed by self-assembly of two or more different peptides. Because OBOC libraries are produced by chemical synthesis, the discovered self-assembled nanomaterials are not limited to canonical amino acids. Various nanomaterials with a huge range of building blocks can be developed. The

screening process is very simple and can be easily automated. Physicochemical conditions such as pH, ionic strength, solvents, temperature, electric current, and magnetic field can also be incorporated into the screening steps. Sequential screening of immobilized OBOC library-beads under various conditions, in conjunction with optical spectral analysis will enable us to discover novel nanomaterial with desirable properties, including stimuli-responsive property that is very important in biomedical applications.” We hope that this discussion will satisfy reviewer’s concern on the usefulness and importance of this highly enabling and yet very simple method.

Regarding the entrance of nanoparticles into HeLa cells, we have carried out additional experiments to address reviewer’s concern. To track the biodistribution of these peptides in the cytosol, peptides YFTEF, ISDNL and FGFDP were labeled by addition of FITC-labeled peptides (FITC-YFTEF, FITC-ISDNL, FITC-FGFDP) at 3% level, respectively, prior to self-assembly, and then incubated with HeLa cells for 2 h. The incubated cells were then washed with PBS three times and further incubated with Dulbecco's modified eagle medium without peptide for another 2 and 6 h. The lysosomes of the cells were stained with red fluorescence after 2 (peptide for 2 h), 4 (peptide for 2 h and DMEM for 2 h), or 8 h (peptide for 2 h and DMEM for 6 h) incubation, followed by CLSM observation. The results are shown in Figure R10.

For ISDNL, good co-localization (Pearson correlation coefficient: 0.92) between peptides and lysosomes was found at 2 h, indicating that the peptide ISDNL based nanoparticles entered lysosome in 2 h. The green fluorescence (4 h and 8 h) in cell gradually increased which was not co-localized with red fluorescence from lysosomes (colocalization from 0.92 at 2 h, to 0.76 at 4 h and 0.69 at 8 h). These results suggest that the ISDNL based nanoparticles could escape from lysosomes over time.

For YFTEF, good co-localization (yellow fluorescence) between YFTEF (green fluorescence) and with lysosomes (red fluorescence) was seen at 2 h (0.92). The co-localization efficiency for YFTEF and lysosomes remained high over time (4 h, 0.93; 8 h, 0.91). Based on these results, we can conclude that the YFTEF based nanoparticles entered lysosomes in 2 h and remained in lysosomes at 8 h without obvious lysosomal escape.

For FGFDP, the Pearson correlation coefficient of lysosome was determined to be about 0.7 for 2 (0.74), 4 (0.79), 8 h (0.72). It seems that FGFDP might have escaped from lysosomes prior to the 2 h time point. In order to confirm this hypothesis, we did the co-localization study at earlier time points: 30 min to 1 h. The results reveal that FGFDP increased the distribution in the lysosome from 30 min (Pearson correlation coefficient: 0.59) to 1 h (Pearson correlation coefficient: 0.94), and then decreased to 0.74 at 2h. These results indicate that the peptide FGFDP nanoparticles did enter lysosomes in the first hour, and began to escape from lysosomes sometime between 1-2 h.

In summary, YFTEF was found to enter the lysosome in 2 h, and remained in lysosomes at 8 h. ISDNL and FGFD entered lysosome and then escaped. The ISDNL escaped in 4 h, and FGFD escaped between 1-2 h. Both ISDNL and FGFD had capability of lysosomal escape, which may be useful for gene or nucleic acid delivery, as implied by the reviewer.

Figure R10. a. Time-dependent CLSM images for monitoring and quantitatively analyzing HeLa cells incubated with 50 μ M pentapeptide nano-assembly of YFTEF, ISDNL, and FGFD, b. corresponding co-localization analysis of experiment shown in “a”.

To confirm how efficient the peptides (YFTEF, ISDNL, and FGFD) enter living cells, HeLa cells were incubated with peptides for 4 h, the cells were then lysed and fluorescence quantified to calculate the cell uptake efficiency. The results are shown in Table R2.

Table R2. Cell uptake efficiency of peptides.

Peptides	YFTEF	ISDNL	FGFD
Efficiency	3.9%	2.2%	4.8%

To elucidate the cell uptake pathways of pentapeptides, we quantified the fluorescence intensity of the HeLa cells after incubation with FITC-labeled nano-assembly of YFTEF, ISDNL, FAGFT, FGFD, at 37 and 4 $^{\circ}$ C, respectively (3% of the peptide assembly were N-terminally labeled with FITC for tracking). There were nearly no uptake of pentapeptides by HeLa cells at 4 $^{\circ}$ C, which is consistent with mechanism of cell uptake by endocytosis. We have also examined the effects of cell uptake by various endocytosis inhibitors: amiloride (2 mM, inhibitor of endocytosis), M- β -CD (5 mM, inhibitor of caveolae-dependent endocytosis), and chlorpromazine (50 μ M, inhibitor of clathrin-dependent endocytosis). As shown in Figure R11 and S23

in supporting information, chlorpromazine was found to suppress the cell uptake of YFTEF, indicating a clathrin dependent endocytosis mechanism. Uptake of ISDNL and FAGFT was significantly decreased by amiloride and M-13-CD, suggesting that ISDNL might enter cells by caveolae-dependent endocytosis and micropinocytosis. For FGFD, chlorpromazine and M-13-CD significantly decreased cell uptake, suggesting both caveolae-dependent and clathrin-dependent endocytosis.

Figure R11. Cell uptake pathway detection. a) CLSM images of HeLa cells incubated with pentapeptides (50 μ M) at different temperatures (37 or 4 $^{\circ}$ C) and in the presence of various endocytosis inhibitors, such as amiloride (2 mM), M-13-CD (5 mM), and chlorpromazine(50 μ M), respectively. The scale bar is 10 μ m. b) Relevant quantitative analysis in 'a'.

Appropriate use of statistics and treatment of uncertainties:

N/A

Suggested improvements:

Since this work is primarily about a new method, it is crucial to demonstrate its "utility" in some significant fashion, e.g., discovery of materials of novel properties or the nanoparticles outperform current systems for cargo delivery.

Response:

We thank reviewer's suggestion. The identified self-assembling peptides aggregate dynamically, initially from nanoparticles and later into nanofibers (Figure 3). This unique dynamic self-assembling property might afford some useful applications, such as delivery of payloads to specific organelles or nucleus inside the cells (Figure 4). We conducted some drug delivery experiments by loading doxorubicin (DOX), a common chemotherapeutic agent, into the peptide nanoparticles, to see if LDFPI (distribution in nucleus) and ISDNL (distribution in cytoplasm) would perform differently in cell culture cytotoxic assay (Figure R12). We found that cells treated with pentapeptide LDFPI loaded DOX showed lower cell viability than that of ISDNL peptide loaded DOX when the concentration of peptide was higher than 50 μM . This could be ascribed to different intracellular biodistribution of LDFPI (distribution in nucleus) and ISDNL (distribution in cytoplasm) nanocarriers. The LDFPI may increase the DOX biodistribution into the nucleus, where DOX can intercalate DNA to exert its cytotoxic function. Much more work will need to be done before any definitive conclusion can be drawn. Nonetheless, this preliminary result does suggest that self-assembling peptides identified by OBOC technology, not only can provide useful data for our understanding of the fundamentals of peptide assembly, but also provide useful information for design of nanocarrier for drug or DNA/RNA delivery.

Figure R12. CCK-8 cell viability assay on HeLa cells after incubation of polypeptides (ISDNL, LDFPI) loaded with DOX for 24 h.

Clarity and context:

The manuscript was clearly written for the most part, but need some improvement in scientific terminologies, especially for the Introduction section. For example, "D-amino acids" are one type of "unnatural amino acids". There is no such thing as the "eukaryotic L-amino acid". "small molecules" include "small-molecule inhibitors". "cyclic peptides" and "macrocytic peptides" generally mean the same thing. "unnatural folders" include "peptoids". These terms were often described as parallel terms.

Response:

We thank reviewer's suggestion. We have revised the manuscript accordingly.

REVIEWER COMMENTS

Reviewer #1 (Remarks to the Author):

The authors have overall addressed my previous comments convincingly. The manuscript has improved and the power of this screening methodology has been convincingly demonstrated. Therefore, the paper is recommended for publication. The following minor comment should however be considered prior to publication:

My previous comment 1, in part, read, "it would be interesting to get a population level insight e.g. on how many of the 100,000 beads that were analyzed are considered 'positive'." I still believe that this is an important message for a paper that aims to provide insights into the self-assembly sequence space by using a combinatorial approach. The authors responded, but the point is not discussed in the paper.

While the authors have now performed the analysis of 536 beads where 2.7% were judged to be sufficiently fluorescent, this was not included in the paper or supplementary information. In fact this now raises a question – how does this finding relate to "From a total of 100,000 beads screened, we selected a total of eight strongest fluorescent beads and four dark beads as negative beads." Please clarify this point.

The authors are encouraged to analyze a few more of these similar size (500 or so) sets to get a better idea of the statistical % hit rate in the overall library. The current analysis suggests that there may be around 70,000 self-assembling peptides in the entire 2.5M library, which is a notable number this is useful context for the reader. Also, can they please clarify how they judge beads to be sufficiently 'bright'? (Figure 1R) Is this a judgement by the microscopist or do they use image analysis to quantify?

Reviewer #2 (Remarks to the Author):

This is an excellent, comprehensive study on a new methodology for the identification of self-assembling peptides from OBOC peptide libraries. Several short peptides have previously been shown to self-assemble into fibrous structures in aqueous solutions. The authors of this manuscript developed a very simple method to rapidly identify additional peptide sequences with self-assembling activities, namely by monitoring the increased fluorescence of NBD following peptide aggregation on beads. This method should be generally useful for discovering other peptides/peptidomimetics that self-assemble into nanostructures. This study also significantly expanded the peptide sequences known to undergo self-assembly. The authors are commended for applying a battery of biophysical techniques to subsequently characterize the structures of the peptide nanostructures formed.

Some very minor changes are recommended:

1) Cellular entry into HeLa cells was demonstrated primarily by confocal microscopy. Treatment with low temperature and endocytosis inhibitors established that the peptide nanoparticles enter the cell by endocytosis mechanisms. However, endosomal escape and cytosolic/nuclear entry have not been adequately established. While some fluorescence signals are visible inside the nuclear region, these signals could be caused by improper selection of the z stack and the overlap of cytosolic and nuclear sections. The authors may either perform additional experiments that more definitively demonstrate cytosolic/nuclear entry or simply tone down their claim of cytosolic/nuclear entry.

2) Please provide more descriptive legends for some of the figures, e.g., what is the peptide signal colocalized with in Figure S23?

3) Please use proper significant figures for all reported values in Table 1 and elsewhere.

4) In Figure 3d, the 1664 cm^{-1} band is hardly visible. Is it significant?

Reviewer #3 (Remarks to the Author):

The authors had fully addressed all my concerns.

Point by point response to the editor's comments

Response to the reviewer #1's comments:

The authors have overall addressed my previous comments convincingly. The manuscript has improved and the power of this screening methodology has been convincingly demonstrated. Therefore, the paper is recommended for publication. The following minor comment should however be considered prior to publication:

My previous comment 1, in part, read, "it would be interesting to get a population level insight e.g. on how many of the 100,000 beads that were analyzed are considered 'positive'." I still believe that this is an important message for a paper that aims to provide insights into the self-assembly sequence space by using a combinatorial approach. The authors responded, but the point is not discussed in the paper.

While the authors have now performed the analysis of 536 beads where 2.7% were judged to be sufficiently fluorescent, this was not included in the paper or supplementary information. In fact this now raises a question – how does this finding relate to "From a total of 100,000 beads screened, we selected a total of eight strongest fluorescent beads and four dark beads as negative beads." Please clarify this point.

The authors are encouraged to analyze a few more of these similar size (500 or so) sets to get a better idea of the statistical % hit rate in the overall library. The current analysis suggests that there may be around 70,000 self-assembling peptides in the entire 2.5M library, which is a notable number this is useful context for the reader. Also, can they please clarify how they judge beads to be sufficiently 'bright'? (Figure 1R) Is this a judgement by the microscopist or do they use image analysis to quantify?

Response:

In response to the reviewer's suggestion, we have performed a more comprehensive screening experiment using approximately 200,000 beads. Beads were immobilized on a planar polystyrene surface using a solvent mixture of dichloromethane (DCM) and dimethylformamide (DMF) at 1:4 ratio (v/v), followed by submersion of the immobilized library beads in water. After equilibration for >30 minutes, the entire plate was then scanned with confocal microscope using an automatic stage, and the stitched fluorescent images analyzed with MATLAB software as previously reported ((Carney, R. P. et al. Combinatorial Library Screening with Liposomes for Discovery of Membrane Active Peptides. ACS Comb. Sci., 19, 299-307 (2017)). Figure 1 shows the fluorescent profile of the entire library with bead number on the Y-axis and relative fluorescent intensity (0-1.0) on the X-axis. It turns out that a total of 186,288 beads were actually screened. Of these 186,288 beads, 79 or ~0.04% displayed a fluorescence intensity ≥ 0.99 ,

and 559 or 0.3% beads displayed a fluorescence intensity ≥ 0.9 . These 559 beads are considered positive beads with high propensity for self-assembling, because they are able to create a hydrophobic microenvironment for fluorescent activation of the tethered NBD dye. We have added this discussion in the manuscript and figure 1 below has been added to the supplemental data as Figure S15.

Figure S15. Fluorescence profile of an entire immobilized random OBOC pentapeptide library N-capped with NBD dye under aqueous condition, with relative fluorescence unit ranging between 0 to 1.0. Of the 186,288 beads immobilized and screened, 79 or $\sim 0.04\%$ beads displayed a relative fluorescence intensity ≥ 0.99 , and 559 or 0.3% beads displayed a relative fluorescence intensity ≥ 0.9 . Each vertical bar depicts the number of beads present within a 0.01 relative fluorescence unit range.

Response to the reviewer #2's comments:

This is an excellent, comprehensive study on a new methodology for the identification of self-assembling peptides from OBOC peptide libraries. Several short peptides have previously been shown to self-assemble into fibrous structures in aqueous solutions. The authors of this manuscript developed a very simple method to rapidly identify additional peptide sequences with self-assembling activities, namely by monitoring the increased fluorescence of NBD following peptide aggregation on beads. This method should be generally useful for discovering other peptides/peptidomimetics that self-assemble into nanostructures. This study also significantly expanded the peptide sequences known to undergo self-assembly. The authors

are commended for applying a battery of biophysical techniques to subsequently characterize the structures of the peptide nanostructures formed.

Some very minor changes are recommended:

1) Cellular entry into HeLa cells was demonstrated primarily by confocal microscopy. Treatment with low temperature and endocytosis inhibitors established that the peptide nanoparticles enter the cell by endocytosis mechanisms. However, endosomal escape and cytosolic/nuclear entry have not been adequately established. While some fluorescence signals are visible inside the nuclear region, these signals could be caused by improper selection of the z stack and the overlap of cytosolic and nuclear sections. The authors may either perform additional experiments that more definitively demonstrate cytosolic/nuclear entry or simply tone down their claim of cytosolic/nuclear entry.

Response:

In response to the reviewer's suggestion, we have tone down the claim of cytosolic/nuclear entry in the text to "It appears that LDFPI nanoparticles might be able to permeate the cells and eventually be localized to the nucleus and perinuclear areas (Figure 4d).".

2) Please provide more descriptive legends for some of the figures, e.g., what is the peptide signal colocalized with in Figure S23?

Response:

In response to the reviewer's suggestion, we have added the more detail descriptions in the figure legends. "Figure S23 (Figure S24 in the revised manuscript). a) Time-dependent (2, 4, 8 h) CLSM images for monitoring and quantitatively analyzing uptake of fluorescent peptide assemblies by HeLa cells. The peptides (50 μ M) including YFTEF, ISDNL, and FGFDP were labeled by FITC with green fluorescence. The lysosomes of HeLa cells were stained with commercial Lyso Tracker Red with red fluorescence. The scale bar is 10 μ m. b) Corresponding colocalization analysis of peptides and lysosomes in images from 'a'."

3) Please use proper significant figures for all reported values in Table 1 and elsewhere.

Response:

In response to the reviewer's suggestion, we have explained how the data in Table 1, as shown below, was obtained.

Table 1. Peptide information screened from OBOC library

Penta-peptide	Fluorescence (beads)	Assembly	CMC (μM)	Charge ^a	GRAVY ^b
FTISD	ON	YES	10.04	-	0.46
ITSVV	ON	YES	14.8	0	2.28
YFTEF	ON	YES	11.4	-	0.02
ISDNL	ON	YES	9.4	-	0.1
LDFPI	ON	YES	8.4	-	1.2
FAGFT	ON	YES	13.9	0	1.26
FGFDP	ON	YES	14.8	-	0.02
FFVDF	ON	YES	14.4	-	1.82
RITWI	OFF	NO		+	0.58
PLVKA	OFF	NO		+	0.86
PFTTR	OFF	NO		+	-0.94
QIMRW	OFF	NO		+	-0.5

^aThe charges for peptides were calculated based on the charges of amino acid. ^bThe grand average of hydrophobicity (GRAVY) for each peptide was obtained by using the ProtParam program available at <<https://web.expasy.org/protparam/>>

4) In Figure 3d, the 1664 cm^{-1} band is hardly visible. Is it significant?

Response:

In response to the reviewer's suggestion, we have magnified the optical spectrum; the peak at 1664 cm^{-1} is now visible. Figure 3d was replaced as shown below.

d

Response to the reviewer #3's comments:

The authors had fully addressed all my concerns.

Response:

We thank reviewer's support and approval.